# Metabolites Produced by the Oral Commensal Bacterium *Corynebacterium durum* Extend the Lifespan of *Caenorhabditis elegans* via SIR-2.1 Overexpression

**DOI:** 10.3390/ijms21062212

**Published:** 2020-03-23

**Authors:** Jun Hyeong Kim, In Hyuk Bang, Yun Jeong Noh, Dae Keun Kim, Eun Ju Bae, In Hyun Hwang

**Affiliations:** 1Department of Pharmacy, Woosuk University, Wanju, Jeonbuk 55338, Korea; hihaho70@naver.com (J.H.K.); dkkim@woosuk.ac.kr (D.K.K.); 2Department of Biochemistry, Chonbuk National University Medical School, Jeonju, Jeonbuk 54896, Korea; bang--c@hanmail.net; 3National Institute of Animal Science, RDA, Wanju, Jeonbuk 55365, Korea; nyj9408@hanmail.net; 4College of Pharmacy, Chonbuk National University, Jeonju, Jeonbuk 54896, Korea; ejbae7@jbnu.ac.kr

**Keywords:** human microbiota, lifespan-extending activities, *Corynebacterium durum*, *Caenorhabditis elegans*, phenethylamine, *N*-acetylphenethylamine, SIR-2.1

## Abstract

Human microbiota is heavily involved in host health, including the aging process. Based on the hypothesis that the human microbiota manipulates host aging via the production of chemical messengers, lifespan-extending activities of the metabolites produced by the oral commensal bacterium *Corynebacterium durum* and derivatives thereof were evaluated using the model organism *Caenorhabditis elegans*. Chemical investigation of the acetone extract of a *C. durum* culture led to the identification of monoamines and *N*-acetyl monoamines as major metabolites. Phenethylamine and *N*-acetylphenethylamine induced a potent and dose-dependent increase of the *C. elegans* lifespan, up to 21.6% and 19.9%, respectively. A mechanistic study revealed that the induction of SIR-2.1, a highly conserved protein associated with the regulation of lifespan, was responsible for the observed increased longevity.

## 1. Introduction

The idea of microbe–host interactions in human ecosystems has been supported by many lines of evidence, all showing that the human microbiota—a group of microbes that colonizes the human body—is intimately involved in human health [1,2]. The recent development of gene sequencing technologies has facilitated studies on the collective genes of the human microbiota (i.e., the human microbiome), enabling the identification of unculturable microbes and thereby expanding the diversity of known human microbiota [2]. Major achievements of human microbiome research are represented by observations of drastic changes in microbial composition between healthy and diseased individuals [2,3]. Accordingly, therapeutic methods that alter the microbial composition have been employed in the clinic, such as fecal microbiota transplantation for patients with autism spectrum disorder and Parkinson’s disease [4,5].

Approximately 700 microbial species are present in the human oral cavity. The human oral microbiota easily migrates to other parts of the body as the oral cavity is the initial gateway to the digestive and respiratory tracts, thereby exerting significant influences on human health [6,7]. In addition to infectious oral diseases, several systemic diseases have been reported to be associated with oral microbiota. A two-way relationship between diabetes and periodontitis has been well established, and the oral microbiota has been suggested to be the link between the two diseases. Distinctive changes in salivary microbial composition have also been found in subjects with obesity and inflammatory bowel disease (IBD), reinforcing the importance of oral microbiota in maintaining health [7,8]. However, the molecular mechanisms underlying the effects of oral microbiota on human health are not fully understood. In particular, chemical investigations of human microbiota have been limited, although metabolites are believed to be chemical messengers mediating the microbe–host interactions [9].

In general, aging is accompanied by changes in human microbial composition toward an imbalanced state with reduced biodiversity (i.e., dysbiosis), a state that has been reported to show strong correlations with various diseases, such as obesity, type 2 diabetes, Alzheimer disease, and Parkinson’s disease [10]. These findings are consistent with the statistics that aging, a well-known risk factor for many diseases, increases the incidences of these diseases. Also, patterns and profiles of human microbiota were recently applied to the development of methods for age prediction [11].

Based on the close correlations of human microbiota with host aging, a representative indicator of host health [10,11], we proposed the hypothesis that the human microbiota generates chemical messengers to control host aging, as a part of microbe–host interactions. *Corynebacterium durum* is a human commensal bacterium that colonizes the oral cavity, as well as the digestive and respiratory tracts, and is reportedly one of the dominant species in healthy colon specimens [12]. In addition, a systemic study on biosynthetic gene clusters (BCGs) of the human microbiome revealed high numbers of BCGs present in the genus *Corynebacterium* [13], implying that *C. durum* is a potential prolific producer of chemical messengers beneficial to human health.

Although the specific microbial species and the proportion thereof vary significantly from person to person, the phylum Actinobacteria, which includes the genus *Corynebacterium*, was reported to be present in the oral cavity at a higher proportion than in the gastrointestinal tract [14]. Moreover, oral microbes are easy to access and culture in the laboratory. In the present study, chemical investigation of *C. durum* collected from human subgingival dental plaque led to the identification of monoamines and *N*-acetyl monoamines as the major components of the culture extract. In addition, inspired by the chemical profile generated by *C. durum*, the anti-aging activities of phenethylamine, tryptamine, and tyramine, as well as their *N*-acetyl derivatives, were evaluated in *Caenorhabditis elegans*, a widely used model organism for human aging studies mainly due to its short average lifespan (about 3 weeks) and completely sequenced genome [15,16].

Several key proteins involved in the regulation of lifespan have been identified, which include the budding yeast Sir2p (silent information regulator 2), a founding member of the sirtuin family, and its ortholog SIR-2.1 in *C. elegans* [17,18]. These Sir2 family members are NAD^+^-dependent histone deacetylases, encoded by the *SIR2* gene, which is highly conserved in organisms ranging from yeast to humans. The mammalian Sir2 homolog is SIRT1, which deacetylates transcription factors and increases the resistance of mammalian cells to DNA damage-induced apoptosis [19,20]. In addition, SIRT1 expression decreases during aging [21]. Despite the importance of Sir2 in animal health, only a limited number of small molecules that regulate Sir2 expression have been reported, which include the plant metabolite resveratrol [22].

The mechanism of the activity observed in this study was found to be the induction of the sirtuin gene *sir-2.1*, which has been shown to play key roles in the regulation of the *C. elegans* lifespan [17,18]. Details of the isolation and structure identification of monoamines and *N*-acetyl monoamines, as well as a mechanistic study of their effects on the *C. elegans* lifespan, are presented here.

## 2. Results

### 2.1. Identification and Synthesis

*N*-acetylphenethylamine (N-APEA), tyramine (TA), tryptamine (TRA), and *N*-acetyltryptamine (N-ATRA) were isolated as the major metabolites produced by the oral commensal bacterium *Corynebacterium durum*. The structures were identified by comparison of ^1^H-NMR data with those of authentic standards and their synthetic *N*-acetylation products and were independently confirmed through analysis of LC-MS data (Figure 1). Although phenethylamine (PEA) and *N*-acetyltyramine (N-ATA) were not detected in the culture extract, these two compounds (PEA and N-ATA) and the four isolates (N-APEA, TA, TRA, N-ATRA) are hypothesized to be chemical messengers for microbe–host interactions because PEA and N-ATA are biosynthetically related to the isolated compounds N-APEA and TA, respectively. Therefore, authenticated samples of PEA, TA, and TRA were prepared and their respective *N*-acetylation products, N-APEA, N-ATA, and N-ATRA, were synthesized. The structures of the synthetic compounds were verified by ^1^H-NMR data analysis.

### 2.2. Lifespan Screening

Lifespan is an important indicator that reflects animal health, and *Caenorhabditis elegans* is a well-established animal model in the study of aging [15,16]. Based on the hypothesis that the human microbiota produces chemical messengers to maintain a healthy host, lifespan screening of the six compounds—PEA, TA, TRA, N-APEA, N-ATA, and N-ATRA—was undertaken using wild-type N2 *C. elegans*. All of the compounds extended the lifespan from 6.6% to 30.4% at a concentration of 50 μM relative to that of the untreated control groups (Figure 2). In particular, PEA and N-APEA showed potent lifespan-extending activity among the monoamines and *N*-acetylmonoamines, respectively.

### 2.3. Lifespan Assay of PEA and N-APEA

Based on the lifespan screening data which displayed the longevity potential of PEA and N-APEA, the two compounds were re-evaluated for their lifespan-extending effects at three different concentrations (25, 50, and 100 μM) using the wild-type N2 worms. Both PEA (Figure 3 and Table 1) and N-APEA (Figure 4 and Table 2) showed concentration-dependent activity on the longevity of *C. elegans*. The estimated mean lifespan of the PEA-treated nematodes was increased by 12.4% (12.5 ± 0.3 days, *p* < 0.001), 19.8% (13.3 ± 0.3 days, *p* < 0.001), and 21.6% (13.6 ± 0.3 days, *p* < 0.001) at 25, 50, and 100 μM, respectively, relative to that of the untreated control group (11.1 ± 0.3 days). In addition, treatment of the worms with N-APEA demonstrated a 7.8% (12.6 ± 0.3 days, *p* < 0.05), 12.6% (13.2 ± 0.3 days, *p* < 0.001), and 19.9% (14.0 ± 0.4 days, *p* < 0.001) increase in the estimated mean lifespan at 25, 50, and 100 μM, respectively, relative to that of the negative control group (11.7 ± 0.3 days). The positive control 4-HBA [23] showed a slightly more potent lifespan-extending activity at 25 μM than that of PEA or N-APEA at the same concentration.

### 2.4. SIR-2.1 Protein

The six compounds analogous to the major metabolites of *C. durum* were inactive in DPPH radical scavenging assays (Appendix A), showing an inconsistency with the biological properties of anti-aging natural products encountered in our former studies [23,24]. Therefore, expression of the proteins known to be involved in the regulation of lifespan, such as the evolutionarily conserved protein SIR-2.1, were hypothesized to be the underlying mechanism of the lifespan extension induced by PEA and N-APEA. The protein expression levels of SIR-2.1 were evaluated by western blot analysis after the cultivation of *C. elegans* with PEA or N-APEA at 50 and 100 µM for 4 days. As a result, the expression of SIR-2.1 was increased by the treatment as shown in Figure 5.

### 2.5. Lifespan Assay in VC199 sir-2.1(ok434)

The mutant strain VC199 *sir-2.1 (ok434∆)*—which produces an incomplete SIR-2.1 protein—was used to confirm that the effect of PEA and N-APEA on lifespan is mediated by increases in SIR-2.1 expression. The western blot analysis verified that the mutant strain does not express SIR-2.1 protein, as shown in Figure 6a. The control group, which consisted of untreated *sir-2.1* mutant worms, showed a shorter mean lifespan (10.5 ± 0.3 days) relative to that of the wildtype N2 nematodes, consistent with the literature reports of this mutant strain [22,25]. Moreover, the lifespan-extending effects of PEA or N-APEA were not observed in the *sir-2.1* mutant worms, indicating that both PEA and N-APEA increase the lifespan of *C. elegans* through SIR-2.1 (Figure 6b).

## 3. Discussion

Based on the strong correlations reported between microbes and the health condition of their host [1,2,3], chemical messengers produced by human microbiota are hypothesized to be the mechanism they use to manipulate human health. *Corynebacterium durum* colonizes the human oral cavity, as well as the digestive and respiratory tracts, and is reportedly one of dominant species in healthy colon specimens [12]. In addition, the genus *Corynebacterium* has been reported to contain high numbers of BCGs [13], suggesting that the human microbes belonging to this genus are prolific producers of chemical messengers.

In the present study, major metabolites produced by *C. durum* were identified, and their potential ability to control host health was investigated using *Caenorhabditis elegans*, a well-established model organism in aging research, as longevity is one of representative health metrics. Four amino acid-derived compounds, N-APEA, TA, TRA, and N-ATRA, were isolated from the acetone extract of the *C. durum* culture via various chromatographic methods, and their structures were identified by analysis of NMR and LC-MS data. Recent studies demonstrated that TRA produced by the human gut microbiota increases fluid secretion in the proximal colon and reduces the production of fatty acid- and lipopolysaccharide-stimulated pro-inflammatory cytokines in macrophages [26,27]. These results suggest that TRA contributes to intestinal and systemic homeostasis in health as a crucial chemical messenger [28,29]. While decarboxylation of tryptophan is rare in bacterial metabolism, about 10% of the human population is estimated to possess at least one gut bacterium that contains genes encoding homologs of the enzyme tryptophan decarboxylase [29].

Inspired by the chemical structures of the major metabolites obtained from *C. durum*, PEA, TA, TRA, and their synthetic *N*-acetylation products were evaluated for lifespan-extending effects in *C. elegans*. Screening of the six compounds at 50 μM using the wild-type *C. elegans* resulted in extension of its lifespan ranging from 6.6% to 30.4% relative to that of the untreated controls. The two most potent compounds revealed by the screening were PEA and N-APEA, both of which showed a dose-dependent increase in the *C. elegans* lifespan at 25, 50, and 100 μM. The maximum increases observed upon treatment with PEA and N-APEA were 21.6 and 19.9%, respectively, at 100 μM. Interestingly, unlike other anti-aging natural products that we have encountered [23,24], the six compounds failed to show DPPH radical scavenging activities (Appendix A).

Acetylation and deacetylation of histones control the accessibility of transcription factors to target genes via conformational changes in DNA, leading to the regulation of a wide range of cellular functions [17,18]. Sirtuins are NAD^+^-dependent histone deacetylases, and are homologs of *Saccharomyces cerevisiae* Sir2p, a founding member of the sirtuin family. Overexpression of Sir2p and homologs thereof, including the metazoan SIR-2.1 and the mammalian SIRT1, has been implicated in the molecular mechanisms of longevity, indicating that sirtuins are highly conserved proteins with therapeutic potential for age-related diseases.

Therefore, the mechanism underlying the longevity caused by PEA and N-APEA was proposed to be associated with the expression of SIR-2.1 in *C. elegans*. The expression level of SIR-2.1 in the wild-type worms was evaluated by western blot analysis after treatment with PEA and N-APEA, showing an increase in the protein levels relative to those of the untreated controls. The proposed mechanism was also supported by the failure of their lifespan-extending effects in the *C. elegans* mutant strain VC199, which does not produce a full-length SIR-2.1 protein. On the contrary, the treatment tended to decrease the mean lifespan of the *sir-2.1* mutant worms. A literature search revealed that PEA causes oxidative stress and inhibits respiration in yeast [30], and sirtuin proteins are known to mediate an oxidative stress response via modulation of antioxidant gene expression [25]. Thus, the SIR-2.1 induction is possibly a part of protective mechanisms against the oxidative stress caused by PEA and N-APEA in *C. elegans*.

Given the biosynthetic relationship between TRA and N-ATRA isolated from the *C. durum* culture extract and the presence of N-APEA therein, PEA was presumed to be produced by the bacterium. Thus, both PEA and N-APEA are hypothesized to be chemical messengers for microbe–host interactions. Because SIR-2.1 is a highly conserved protein involved in the regulation of the longevity of *C. elegans*, the induction of SIR-2.1 expression by PEA and N-APEA suggests that they have potential for improving mammalian health and longevity. These results reinforce the concept that the human microbiota influences host health via production of chemical messengers.

## 4. Materials and Methods

### 4.1. General Experimental Procedures

Optical density was measured at 600 nm with an ELISA microplate reader (Tecan Sunrise, Grödig, Austria). Flash chromatography was carried out using a CombiFlash Retrieve system, equipped with RediSep prepacked columns (Teledyne Isco, Lincoln, NE, USA). HPLC separations employed either a Younglin Acme 9000 system (Seoul, South Korea) connected to a semi-preparative Gemini C18 (Phenomenex, 5 μm, 1.0 × 25 cm) column or a Gilson system (Gilson Medical Electronics, Middleton, WI, USA), which consists of Gilson 305 and 306 pumps and a Gilson UV/VIS 151 detector, with a preparative Luna C18 (Phenomenex, 15 μm, 2.1 × 25 cm) column. NMR spectra were obtained from JEOL JNM-ECZ500R or -ECZ600R spectrometers (Tokyo, Japan). Low-resolution LC/ESIMS data were recorded using an Agilent 6410 Triple Quadrupole LC/MS system with a Zorbax Eclipse SBD-C18 (Agilent, 3.5 μm, 0.21 × 15 cm) column at a flow rate of 0.5 mL/min. Tryptamine, tyrosine, phenethylamine, and acetic anhydride were purchased from Sigma-Aldrich (St. Louis, MO, USA). Brain heart infusion was purchased from BD Biosciences (San Jose, CA, USA).

### 4.2. Bacterial Source and Extraction

*Corynebacterium durum* ChDC OS33, isolated from human subgingival dental plaques using procedures that have been previously described [31], was provided by the Korean Collection of Oral Microbiology (Accession number: KCOM 1047). A culture of the isolate was also deposited at the Korean Collection for Type Cultures under the accession number KCTC 19318. Partial 16S rRNA gene sequence analysis enabled its taxonomic assignment, using polymerase chain reactions with sequencing primers and a nucleotide-to-nucleotide BLAST query of the GenBank database. The sequence information was deposited in GenBank with the accession number AF543285. The oral bacterial strain *C. durum* ChDC OS33 was cultured in nine 2 L Erlenmeyer flasks, each containing 1 L of BHI medium and shaken at 200 rpm at 37 °C. After four weeks of incubation, sterilized XAD-7-HP resin (20 g/L) was added to the culture, and the mixture was shaken at 200 rpm for 2 h to adsorb the organic materials. The resulting resin was collected by filtration through cheesecloth, washed with deionized water, and extracted with acetone [32]. The acetone-soluble portion was evaporated under vacuum to yield 14.7 g of a brown residue.

### 4.3. Purification

The acetone extract (14.7 g) was partitioned between CHCl_3_ (3 × 270 mL) and H_2_O (270 mL). The organic solvent-soluble fraction was dried under vacuum (1.6 g) and then partitioned between hexane (3 × 15 mL) and MeCN (15 mL) to remove lipids. The MeCN-soluble portion (1.3 g) was fractionated by flash column chromatography on silica gel (4.0 × 20 cm, 80 g), eluting with a stepwise gradient of hexane/EtOAc (400 mL each of 1:0, 8:2, 6:4, 4:6, 2:8, 0:1) and EtOAc/MeOH (400 mL each of 6:4, 4:6), followed by pure MeOH (1 L), to afford *N*-acetylphenethylamine (81.4 mg; eluted as Fr.6 with hexane/EtOAc 0:1) and nine other fractions (Fr.1—Fr.5 and Fr.7—Fr.10). Fr.7 (20 mg out of 690 mg) was subjected to C_18_ HPLC (15 μm, 25 × 2.1 cm, 5 mL/min) with 30% MeCN/H_2_O for 5 min, followed by a linear gradient to 75% over 30 min to give *N*-acetyltryptamine (1.1 mg, *t*_R_ = 29.7 min). Tryptamine (12.2 mg; *t*_R_ = 15.5 min) and tyramine (1.9 mg; *t*_R_ = 6.9 min) were obtained from Fr.10 (34 mg out of 155 mg) by C_18_ HPLC (5 μm, 25 × 1.0 cm, 2 mL/min) using 10% MeOH/H_2_O (0.1% formic acid) for 5 min, followed by a linear gradient to 55% over 15 min. Also, the presence of *N*-acetylphenethylamine was detected in Fr.5, and that of tryptamine was found in Fr.8 and Fr.9, via analysis of the respective NMR and LC-MS data.

### 4.4. Compounds Synthesis

Acetic anhydride (153 mg, 1.5 mmol) [33] was added to a sample of phenethylamine (121 mg, 1.0 mmol) dissolved in ethyl acetate (6 mL) in a 20 mL screw-cap glass vial. The reaction vial was sealed with parafilm and then stirred at room temperature for 5 h. The reaction mixture was purified by silica gel flash column chromatography (2.0 × 8.0 cm, 12 g), eluting with EtOAc:MeOH 99:1 to afford *N*-acetylphenethylamine (110 mg). The same procedure was applied to the synthesis of *N*-acetyltryptamine and *N*-acetyltyramine using tryptamine and tyramine, respectively, except that only one equivalent acetic anhydride was added to the sample of tyramine in MeCN.

### 4.5. C. Elegans strains and Maintenance

*Caenorhabditis elegans* strains, including wild-type N2 (var. Bristol) and VC199 *sir-2.1 (ok434)*, and *Escherichia coli* OP50, were obtained from the Caenorhabditis Genetic Center (CGC; University of Minnesota, Minneapolis, MN). Worms were grown and maintained under standard conditions at 20 °C on nematode growth medium (NGM; 2% (*w/v*) agar, 0.3% (*w/v*) NaCl, 0.25% (*w/v*) peptone, 1 mM CaCl_2_, 5 mg/mL cholesterol, 25 mM KH_2_PO_4_, 1 mM MgSO_4_) agar plates seeded with *E. coli* OP50 (OD600 ≈ 0.7) [34].

### 4.6. Lifespan Assay

Lifespan assays were carried out utilizing synchronized L4-stage to young adult-stage worms, which were obtained by the hypochlorite treatment of gravid adults and subsequent cultivation of the resulting embryos at 20 °C [35]. Age-synchronized nematodes were transferred to fresh NGM plates every alternate day to remove progeny, while dead worms—defined as those that failed to respond to a gentle touch with a platinum wire—were counted daily. Accidental deaths or lost animals were excluded. Each group for the lifespan assays used 50 worms, and each assay was performed three times independently, except for the screening assay. All of the compounds used in this study were dissolved in a DMSO stock solution, and the samples containing each compound in DMSO were added to sterilized NGM at 50 °C to produce a final concentration of 0.1% DMSO (*v/v*).

### 4.7. Western Blot

Embryos collected by treating egg-laying adults with alkaline hypochlorite solution were cultured on NGM plates containing PEA or N-APEA at 20 °C to provide synchronized adult nematodes (L4 + 24 h). The resulting worms were washed with M9 buffer (0.5% NaCl, 0.6% Na_2_HPO_4_, 0.3% KH_2_PO_4_, 0.1% NH_4_Cl) and homogenized [36]. The worm homogenates (20 μg) were separated by 10% SDS-PAGE, and then transferred to PVDF membranes. After blocking the membranes with 5% skim milk to prevent nonspecific antibody binding, the blot was probed with primary antibodies against SIR-2.1 or α -tubulin (YL1/2) (NOVUS Biologicals, CO, USA). The membranes were incubated with conjugating secondary antibodies, and the ECL signals were detected using a Las-4000 imager (GE Healthcare Life Science, Pittsburgh, PA, USA). All experiments were carried out in triplicate. Blots were quantified by densitometric analysis using ImageJ software.

### 4.8. Data Analysis

The Kaplan–Meier method was used to plot the data obtained from the lifespan assays, and the significance of differences between survival curves was assessed by the log-rank test. All error bars represent the mean ± standard error of the mean (SEM) of three independent experiments. Statistical significance of the differences between treated and control groups was determined by one-way analysis of variance (ANOVA). All *p* values < 0.05 were considered to be significant. **p* < 0.05, ** *p* < 0.01 and *** *p* < 0.001 unless stated otherwise.

## Figures and Tables

**Figure 1 ijms-21-02212-f001:**
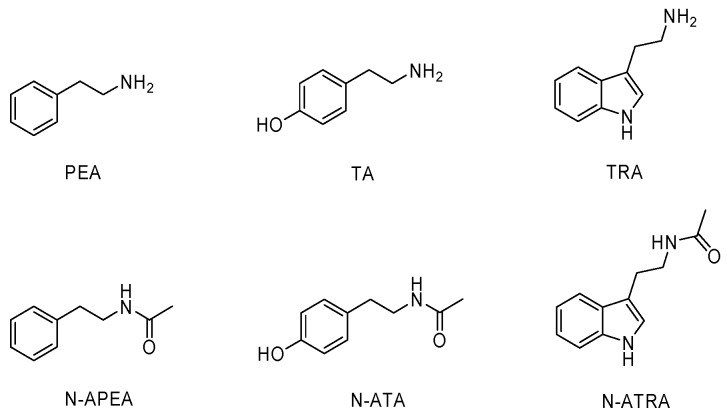
Structures of phenethylamine (PEA), tyramine (TA), and tryptamine (TRA) and their *N*-acetyl derivatives, N-acetylphenethylamine (N-APEA), N-acetyltyramine (N-ATA), and N-acetyltryptamine (N-ATRA).

**Figure 2 ijms-21-02212-f002:**
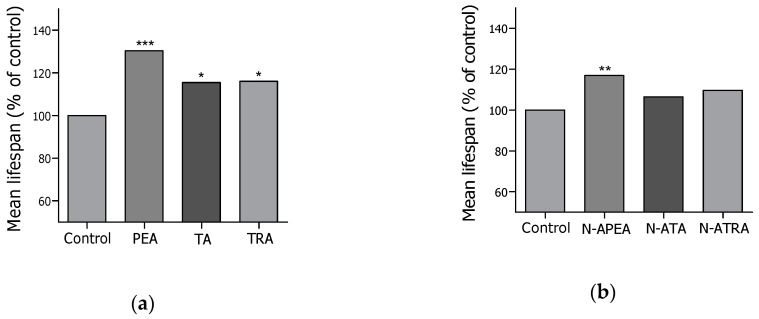
Lifespan screening of PEA, TA, and TRA (**a**), as well as their *N*-acetyl derivatives, N-APEA, N-ATA, and N-ATRA (**b**), in wild-type N2 *C. elegans*. Each screening assay was performed once, and each group in the experiment consisted of 50 worms.

**Figure 3 ijms-21-02212-f003:**
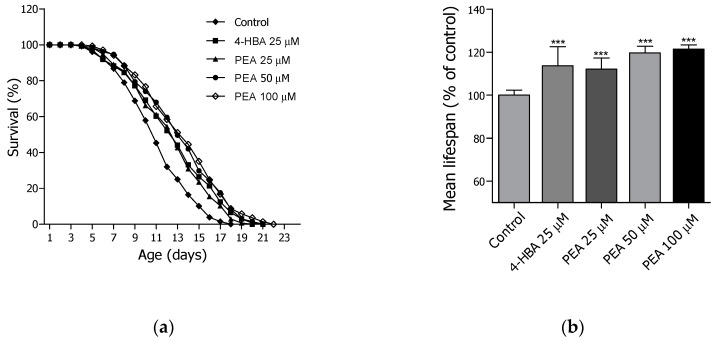
Longevity effects of PEA in wild-type N2 *C. elegans* at 25, 50, and 100 µM. (**a**) The mortality of each group was determined daily by counting the dead animals. (**b**) The mean lifespan of the worms was calculated from the survival curves in (a). The bar graph represents the mean ± SEM of three independent experiments, and each group in the experiment consisted of 50 worms. Significance was determined by one-way ANOVA (*** *p* < 0.001). 4-HBA (4-hydroxybenzoic acid) was used as a positive control.

**Figure 4 ijms-21-02212-f004:**
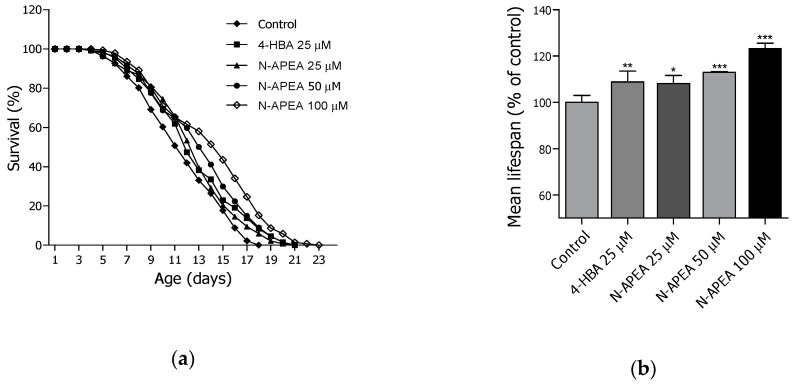
Longevity effects of N-APEA in wild-type N2 *C. elegans* at 25, 50, and 100 µM. (**a**) The mortality of each group was determined daily by counting the dead worms. (**b**) The mean lifespan of the nematodes was evaluated based on the survival curves in (a). The bar graph represents the mean ± SEM of three independent experiments, and each group in the experiment consisted of 50 worms. Significance was determined by one-way ANOVA (* *p* < 0.05, ** *p* < 0.01, and *** *p* < 0.001). 4-HBA (4-hydroxybenzoic acid) was used as a positive control.

**Figure 5 ijms-21-02212-f005:**
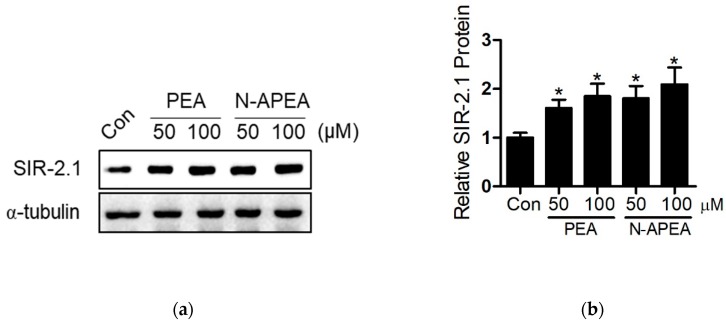
Western blot analysis of SIR-2.1 protein expression in wild-type N2 *C. elegans* upon treatment with PEA or N-APEA at 50 and 100 µM. (**a**) Representative western blot images. (**b**) Quantitative analysis for relative expression of SIR-2.1. The results represent the mean ± SEM of three independent experiments. Significance was determined by one-way ANOVA (* *p* < 0.05).

**Figure 6 ijms-21-02212-f006:**
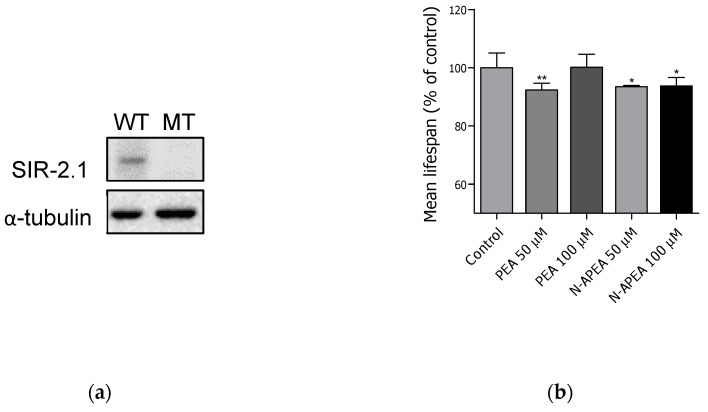
Effects of PEA and N-APEA in *sir-2.1* mutant worms. (**a**) SIR-2.1 protein expression in wildtype (WT) and *sir-2.1* mutant (MT) nematodes. (**b**) The graph displays the mean ± SEM of three independent experiments, and each group in the experiment consisted of 50 worms. The control is the group of untreated *sir-2.1* mutant worms. Differences relative to the control were considered significant at * *p* < 0.05 and ** *p* < 0.01.

**Table 1 ijms-21-02212-t001:** Effects of PEA on the lifespan of *C. elegans.*

Treatment	Dose (µM)	Mean Lifespan^1^ (Day)	Maximum Lifespan (day)	Change in Mean Lifespan^2^ (%)	Log-Rank Test
Control	-	11.1 ± 0.3	18	/	/
4-HBA	25	12.7 ± 0.3	21	13.8	*** *p* < 0.001
PEA	25	12.5 ± 0.3	20	12.4	*** *p* < 0.001
	50	13.3 ± 0.3	21	19.8	*** *p* < 0.001
	100	13.6 ± 0.3	22	21.6	*** *p* < 0.001

^1^ Mean lifespan data are presented as mean ± SEM. ^2^ The change in mean lifespan was calculated relative to the control group (%). Statistical significance of the difference between survival curves was determined by the log-rank test using Kaplan–Meier survival analysis. Differences relative to the control were considered significant at *** *p* < 0.001.

**Table 2 ijms-21-02212-t002:** Effects of N-APEA on the lifespan of *C. elegans.*

Treatment	Dose (µM)	Mean Lifespan^1^ (day)	Maximum Lifespan (day)	Change in Mean Lifespan (%)	Log-Rank Test
Control	-	11.7 ± 0.3	18	-	-
4-HBA	25	12.7 ± 0.3	21	8.3	** *p* < 0.01
N-APEA	25	12.6 ± 0.3	21	7.8	* *p* < 0.05
	50	13.2 ± 0.3	21	12.6	*** *p* < 0.001
	100	14.0 ± 0.4	23	19.9	*** *p* < 0.001

^1^ Mean lifespan data are presented as mean ± SEM. ^2^ The change in mean lifespan was calculated relative to the control group (%). Statistical significance of the difference between survival curves was assessed by the log-rank test using Kaplan–Meier survival analysis. Differences relative to the control were considered significant at * *p* < 0.05, ** *p* < 0.01, and *** *p* < 0.001.

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
