# Peer review of "Metabolites Produced by the Oral Commensal Bacterium Corynebacterium durum Extend the Lifespan of Caenorhabditis elegans via SIR-2.1 Overexpression"

_ijms, 2020, doi:10.3390/ijms21062212_

Round 1

Reviewer 1 Report

To whom is may concern,

The manuscript titled “Metabolites produced by the oral commensal bacterium Corynebacterium durum extend the lifespan of Caenorhabditis elegans via SIR-2.1 overexpression” isolates metabolites from C. durum, tests their effects on the lifespan of C. elegans, and then finds evidence suggesting that a C. elegans protein (SIR-2.1) is upregulated by these metabolites to extend lifespan. I believe this has some interesting ideas and presents data that could ultimately be valuable to the field; however, a number issues need to be addressed before I can recommend the paper be published.

-Overall, I think the language of the manuscript could be improved a little. There were some points where it was unclear what was being stated.

-In the introduction, it would be clearer if the authors provided some rationale for why C. durum being present in the oral cavity is more relevant than the gut and respiratory—is it because it is present in the oral cavity in higher numbers? Is this where the metabolites that effect longevity are thought to be formed (the environments of the oral cavity and gut are markedly different and would presumably result in different metabolite levels/profiles.

-Similarly, why were the culture conditions for C. durum chosen? Growing 1 L worth of bacteria in (presumably aerobic) conditions for 4 weeks seems like a lot. Also, it was stated 14.7 g of a residue was recovered after extraction—what was the volume (in other words, what degree of concentration was performed)?

-Also in the introduction, the authors could improve the manuscript by more explicitly stating what the research gap in knowledge they are addressing and what novelty this experiment has. They do a good job describing how others have isolated and characterized metabolites of C. durum, so where does their study fall if it has already been done? Are the ones discovered here novel? Was the mechanism just not known? This may just be an issue with phrasing—particularly in lines 73-75: was that referring to the results of this study or a previous study? If this study, please remove from the introduction.

-Was PCR or sequencing performed of the sir-2.1 mutant to confirm that the protein was indeed truncated and not functional? If so, please state it. If not, please justify why this was not in the Discussion.

-In Figure 5, I think it would improve the power of the conclusion that the effect was dose-dependent if the authors did some form of semi-quantitative analysis of the figure. Specifically, I recommend analyzing the images from each of the blots for band intensity with ImageJ and normalizing them to the tubulin control.

-In Figure 6, please describe what the control is in the legend—the way I read the text the control was untreated sir-2.1 mutant. Relative to the untreated, it appeared that PEA and n-APEA seemed to actually reduce life span (apparently significantly) relative to the control. What do the authors think the mechanism for this is? It would be good to add to the discussion section. Or was the control the wild type strain? Please clarify.

-In the data analysis section and throughout the assays described, please describe the number of replicates for each sample and experiment. Please do this throughout.

-Lines 93-98 are a little unclear, please rephrase.

Reviewer 2 Report

The idea of the article is original.

The experimental design seems appropriate although Methods references should be given. Actually, too few references are used.

Moreover, a deep English form revision should be performed. 

Author Response

Response to Reviewer 2 Comments
The idea of the article is original.
The experimental design seems appropriate although Methods references should be given. Actually, too few references are used.
Additional references were added in the method section for any possible assistance to the readers and researchers in the related field.
Moreover, a deep English form revision should be performed.
English has been revised, including the sentences pointed out by the reviewers, with a help of English language editing service. We hope that English in the new version of the manuscript is satisfactory to be accepted.

Round 2

Reviewer 1 Report

I thank the authors for thoroughly considering and addressing my points! I believe another read through to improve the clarity of the phrasing in some spots. Overall, it is much improved and I recommend it for publication after copy editing!